# Characterization of Respirable Dust Generated from Full Scale Cutting Tests in Limestone with Conical Picks at Three Stages of Wear

**Syd Slouka** [1,*] , **Jürgen Brune** [1] , **Jamal Rostami** [1] , **Candace Tsai** [2] and **Evan Sidrow** [3]

[1] Colorado School of Mines, Golden, CO 80401, USA; jbrune@mines.edu (J.B.); rostami@mines.edu (J.R.)
[2] School of Public Health, University of California Los Angeles, Los Angeles, CA 90095, USA; candacetsai@ucla.edu
[3] Department of Statistics, University of British Columbia, Vancouver, BC V6T 1Z4, Canada; evan.sidrow@stat.ubc.ca
[*] Correspondence: sslouka@mines.edu

**Abstract:** Respirable rock dust poses serious long-term health complications to workers in environments where mechanical rock excavation is utilized. The purpose of this study is to characterize respirable dust generated by cutting limestone with new, partially worn, and fully worn conical pick wears. Characterizing limestone respirable dust can aid in decision making for respirable dust suppression levels and exposures throughout the lifetime of a pick in underground mining and engineering activities. The methods include full scale cutting of a limestone sample in the laboratory with three conical picks at different stages of wear. Dust samples were collected during cutting with various instruments connected to pumps and subsequently analyzed to determine the concentrations, mineralogy, particle shapes, and particle size distributions. The results show that the worn pick generated the highest concentration of dust, all picks generated dust containing quartz, all three picks generated dust particles of similar shapes, and all three picks generated various particle size distributions. In conclusion, a preliminary suite of respirable dust characteristics is available and with further future additional studies, results could be used for the evaluation of possible strategies and methods of dust suppression and exposures during mining, tunneling, or drilling activities.

**Keywords:** respirable rock dust; conical pick wear; limestone characterization

## 1. Introduction

Airborne rock dust particles pose a major respiratory health concern for workers in underground mining and civil environments. Exposure to rock dust particles less than 4 μm in aerodynamic diameter, or respirable particles, containing silica, coal, or other minerals, can cause irreversible diseases such as coal workers pneumoconiosis [1], silicosis [2] and other lung complications [3,4]. Therefore, the US Mine Safety and Health Administration (MSHA) recognized and addressed these major issues in the 1970s by instating regulations to limit the concentration exposure and exposure to silica. However, even with these regulations, there has been an unfortunate increase in lung disease cases in the United States since the 1990s with a continued rise in numbers in recent years [5–9].

It is not fully understood why there is an increase in modern miners' lung diseases with the concentration and silica exposure limits instated [10,11]. Therefore, various research projects are investigating additional impacts such as analyzing the presence of nanoparticles, particle size distributions, mineralization analysis, and how particle shape affects deposition into the human lung [12–16]. There is also limited understanding of how dust characteristics change as a pick wears down during its lifetime of cutting rock.

The most dominant form of rock excavation in soft and moderate strength rock is the use of mechanized units such as roadheaders, continuous miners, drum shearer, or

similar machines. The primary cutting tools in these machines are conical picks. Dust is generated at the contact point between the pick and rock surface, while additional dust can be generated in the muck handling process. Conical picks generate dust at the carbide tip because they apply concentrated loads to penetrate the rock surface and break the rock into fragments. Pick tip geometry also changes over time as they wear down [17,18]. However, it is uncertain if the pick wear drastically alters dust characteristics as the carbide strikes the rock surface. The only evidence reveals that dust concentration changes with the pick tip geometry and wear of pick tips, but there are no further investigations [17,19–23], nor is there a clear quantitative relationship between the pick tip geometry and the amount of dust generated.

Therefore, this research focuses on the concentration, mineral composition, particle shapes, and particle size distributions of suspended and deposited respirable dust generated from three different conical pick wears during laboratory full scale cutting of a limestone block. A conical pick is used for experimentation because they are commonly used in rock cutting operations for excavation of limestone in a variety of applications [24–27]. The main aim of the work is to characterize the dust particles generated from cutting with different pick tip geometries as it wears out during operation. This can allow the operators to adopt a strategy for bit management along with the consideration of production rate, machine utilization, and consideration of dust suppression measures when cutting limestone in underground metal and non-metal excavation projects.

The highlighted the principal conclusions include the worn pick generating the most overall suspended respirable dust and all three picks generating dust containing quartz with traces of cristobalite. In general, all the picks generated respirable suspended and deposited particles with similar particle shapes in terms of roundness, aspect ratio, and roughness measures. The particle size distributions reveal that the suspended particles and deposited particles generated are all within the respirable size range and have respective trends in relation to pick wear.

## 2. Materials and Methods

### 2.1. Sample Preparation and Full-Scale Cutting

An Indiana limestone rock block was used for experimentation with a Linear Cutting Machine (LCM) at the Earth Mechanics Institute (EMI) at the Colorado School of Mines (CSM). The sample did not contain any apparent joint sets or discontinuities in structure, which provided the most uniform and homogeneous sample possible for consistent cutting tests. Additionally, rock strength properties of the sample were measured in the rock mechanics lab at CSM where the unconfined compressive strength (UCS) was 43.4 MPa, the Brazilian tensile strength was 3.0 MPa, the Cerchar abraisivity index was 0.57, and punch penetration energy slope index was 9.4 kN/mm.

Additionally, a petrography analysis was performed on the limestone sample tested from two core samples extracted. The following observations reveal that the limestone rock was not completely uniform throughout the sample at the microscopic level. For example, the second core analyzed contained an irregular layer of inclined alignment of bioclasts with less interstitial calcite. A diverse assortment of disarticulated brachiopod, echinoderm, calcareous algae, foraminifera, charophyte, and bryozoan were found in both the samples analyzed. Lastly, subangular quartz was found in the inclusion-rich fragments as seen in clastic, micritic Limestone in Figure 1.

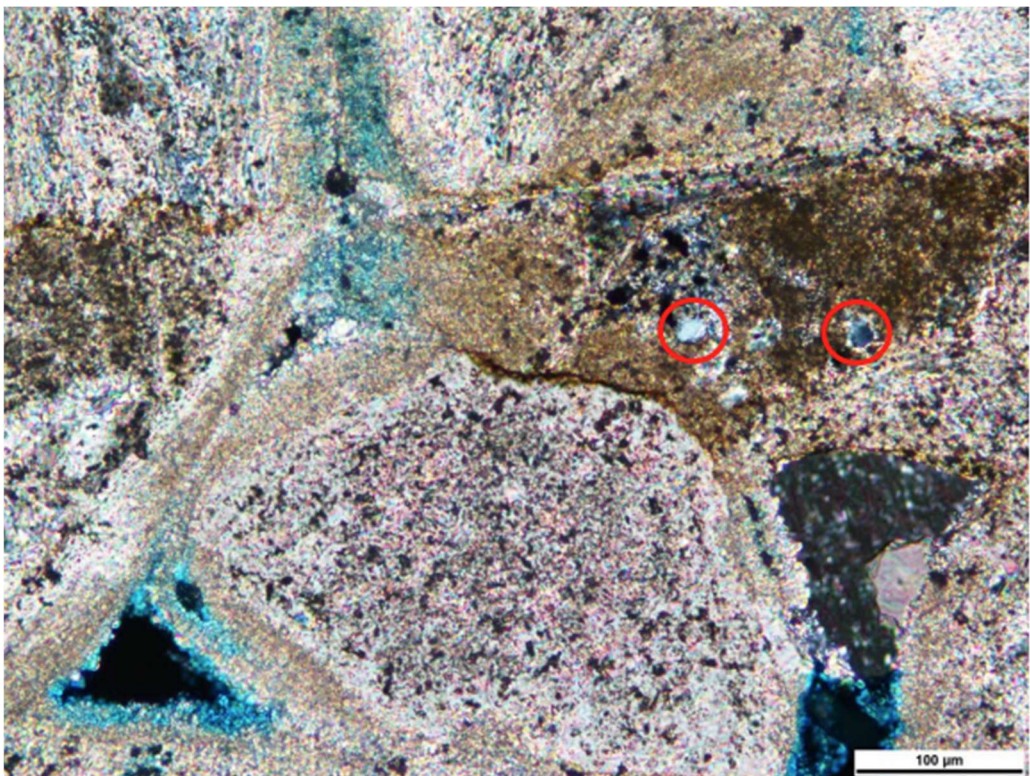

**Figure 1.** Field of view: 0.62 mm, cross-polarized limestone sample. Very-fine grained quartz is included in inclusion-rich fragments. Holes are highlighted by the blue epoxy used in thin section preparation. Red circles reveal smaller grains closer to the respirable size range.

The limestone block was cast into the LCM steel sample box with concrete and placed onto the LCM sled. The sled moves the rock box linearly such that the pick contacts and cuts the sample's surface at a constant speed of 250 mm/sec (10 in/s) and in a linear manner to simulate rock cutting in a full scale rock cutting process. The length of each cut line is about 1.1 m (3.5 ft) and therefore, each cut line takes 4.2 s. Multiple lines of cuts were made across the limestone's surface as seen in Figure 2 at 3.81 cm (1.5-inch) spacings, which is a typical cut spacing for a machine cutting a similar rock with a UCS of 43 MPa. The group of cut lines made horizontally across the surface of the limestone block at the same elevation create one pass across the limestone surface where dust is generated and collected. Each dust sample is collected while cutting all the horizontal lines for one pass at the desired spacing. Additionally, a test set is the dust collected during one pass at the same spacing and penetration for the each of the three differing pick wears.

The penetration, or indentation depth, of the pick into the rock sample was 0.76 cm (0.3 inches) for the experiments. These parameters were used because they are representative of typical spacings and penetrations used for the specific rock type in similar operations and are determined by optimizing the specific energy and normal forces when cutting various rocks [28]. The penetration down into the rock sample was fixed by inserting steel plates between the cross frame and the machine's main structure. For a 5 mm (0.2 in) penetration depth, a 5 mm plate would be used to assure the accuracy of the measured penetration. The plates allow stiffness in the cutting unit since the vertical load is spread over a large area provided by the metal sheet. For penetration values other than 5 mm, multiple of the available 5 mm sheets, or inserts with appropriate thickness (i.e., 2.5, 10, 12.5 mm), can be used.

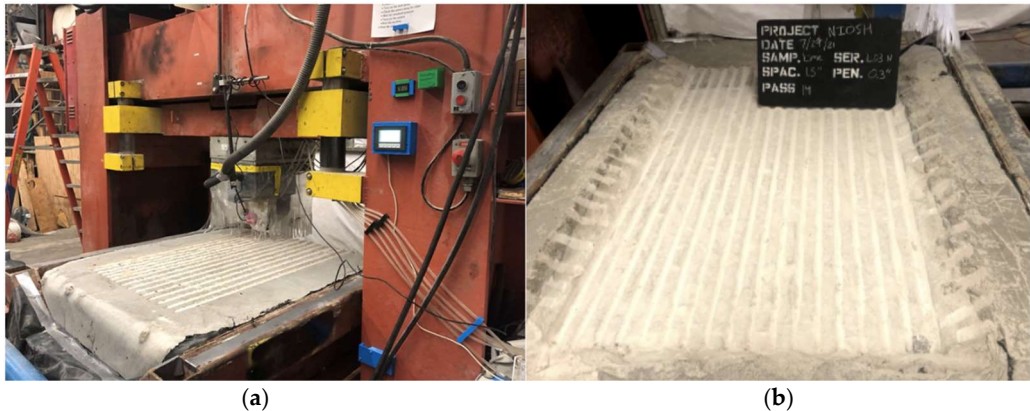

|   |   |
|:-:|:-:|
| (**a**) | (**b**) |

**Figure 2.** Images of (**a**) LCM with the limestone block installed and dust curtain around the pick; (**b**) The surface of the limestone block surrounded by concrete after the completion of a pass at 1.5-inch spacing across the surface.

Using this setup, the LCM ensures constant cutting speed during the cuts with a linear variable displacement transducer (LVDT) sensor. The sensor measures the location of the rock box, or the sample displacement, in real time during testing and the speed can be validated after each cut. The LVDT can be used to locate the pick tip and to separate data when cutting in concrete versus cutting in the target area, in this case the limestone sample.

### 2.2. Dust Collection Setup

Two collection methods were used to obtain representative samples of dust generated from cutting the limestone block. The first method uses nylon 10-mm Dorr-Oliver cyclones with a 50 percent cut point of 4μm aerodynamic diameter particles and a Tsai Diffusion Sampler (TDS) with cutoff aerodynamic diameter of 3.8 μm particles to collect the suspended, or airborne, respirable particles [29]. The second method uses a vacuum (shop-vac®) to collect the particles left behind along the line of cut, or the crushed zone under the bit, on the surface of the rock after each cut. These particles are expected to be on the surface of the rock block after cutting because they have a shorter residence time in the air after cutting and settle onto the rock surface. These deposited particles were collected with the vacuum and studied because they would be transported into downstream operations and potentially re-introduced into the air if the cuts were performed in a real mine operation. Deposited particles were sieved through the US 200 Mesh (<74 μm), where particles passing through the mesh were analyzed.

The Dorr-Oliver cyclones removed larger suspended particles and deposited the respirable-sized particles onto 37-mm diameter polycarbonate (PC) filters with 200 nm pore size and 37-mm diameter PVC filters with 5.0 μm pore size. The TDS collection sampler collected nanometer and suspended respirable sized particles and directly deposited them onto a 25-mm diameter PC filter with 200 nm pore size.

Three cyclones and one TDS inside the dust curtain were equidistant and evenly spaced around the pick to collect dust with the various filters. Polycarbonate filters were used because the substrate is best suited to analyze particles in a scanning electron microscope [30–32]. PVC filters are used to obtain the concentration and mineralogy because the material has a stable weight and high collection efficiency [33–36]. The cyclones and TDS were connected to air pumps with Tygon® tubing. The pumps connected to the cyclones ran at 1.7 L per minute, which is the recommended flow [37], and the pump connected to the TDS ran at 1.0 L per min, which is the recommended flow rate for short-term sampling with high concentration [29]. Due to humidity altering the collection efficiency of cyclones, tests were conducted within a range that does not affect collection efficiency, which was between 20% and 30% humidity [38–41].

The cyclones, TDS, and pumps were integrated into an automated dust collection system as seen in Figure 3. This system eliminates human error in sampling suspended

dust particles during full-scale cutting tests and additionally clears out dust from prior cuts to flush out the system with fresh air from a HEPA filter. With a laser measuring the movement of the rock box, the data interface uses the laser readings to open and close the electric ball valves, as well as turn the flushing system on and off. The automated dust collection system ensures consistency and, therefore, more reliable comparability, between dusts generated with the various picks. Additionally, a rotameter is used to verify pump flow rates before and after cutting and a real-time concentration monitor is used to ensure the flush system removes all particles in the dust curtain before the next cut begins.

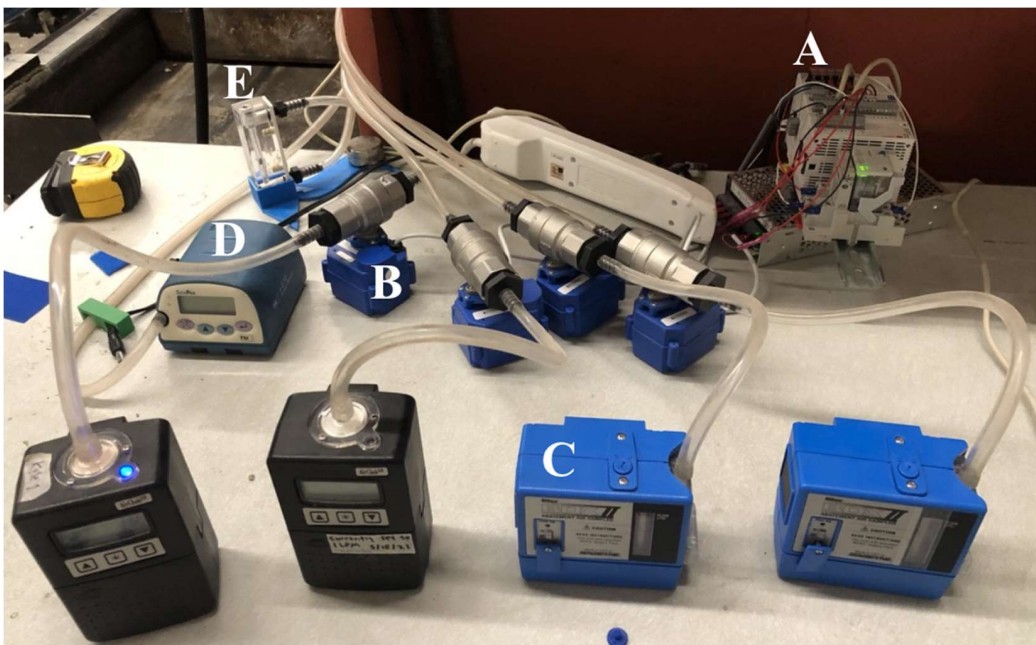

**Figure 3.** (**A**) Data acquisition interface and power unit that controls when dust is collected and flushed out of the system; (**B**) Electric ball valves to control the on and off of air flow through the cyclones and TDS; (**C**) Pumps running at 1.7 or 1.0 L per minute; (**D**) Real-time dust concentration monitor, (**E**) Rotameter currently placed in-line before testing to ensure pumps are running at appropriate flow rate.

Lastly, a curtain was installed around the LCM pick to confine the dust generated from cutting and to keep other unrepresentative lab dusts from being collected. Figure 2 shows the dust curtain around the pick with the fringed bottom to smoothly run along the surface of the limestone sample. The cyclones and TDS instruments were inside the dust curtain and are piped out to the pumps with Tygon® tubing. Before testing, air passing through the HEPA filter is flushed into the dust curtain to remove any environmental air possibly containing particles. After the real-time dust concentration monitor ensures that the concentration of any background emissions has been removed from inside the curtain, testing commences.

### 2.3. Pick Wear Measurement

U92 is a conical pick manufactured by Kennametal Inc with a 105 mm (4 in) gage. This pick type was used for cutting the limestone with a new, moderately worn (assumed mid-life), and worn (assumed end-life) pick. As seen in Figure 4, conical picks have a tip cone angle and a rounded nose with various radii. A circle is superimposed at each of the pick tips to show the rounded nose. The new pick tip has a radius of 1.8 mm (0.07 in), the moderately worn pick has a radius of 3.7 mm (0.15 in), and the worn pick tip has a radius of 5.2 mm (0.2 in). Radius of the tip is used to quantitively represent pick wear because the tip cone angle will stay the same throughout cutting and only the tip itself will become more

blunt over time. The moderately worn and fully worn picks were generated by wearing down a duplicate new pick tip by hand with a Dremel® and a lathe.

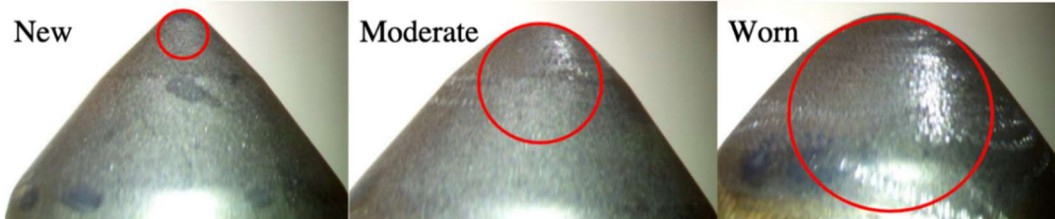

**Figure 4.** Comparison of a new, moderately worn, and worn pick tip with 0.7× magnification. A circle is superimposed to show the increase in pick tip radius with increase in wear.

### 2.4. Particle Characterization

The suspended respirable particles less than 10 μm in diameter are characterized in terms of concentration, mineral content, particle shape, and particle size distribution. Deposited particles, which were the particles left behind on the surface on the sample and sieved through the US 200 Mesh (<74 μm), were characterized in terms of particle shape and particle size distribution. Deposited particles in this paper refers to the particles less than 10μm in diameter that were either originally airborne and settled onto the rock surface, or were particles that remained on the surface after cutting and crushing. Figure 5 shows the purpose of each instrument used for dust collection with the characterization output obtained for the various particles collected. With limited limestone rock sample to cut, the average of three test duplicates were performed for the results obtained.

| Particle Size | Suspended respirable particles (Direct threat) | | | | Deposited particles (indirect threat) |
|---|---|---|---|---|---|
| **Collection Method** | TDS with PC filter | Cyclone with PVC filter | Cyclone with PC filter | Cyclone with PC filter | Vacuum |
| **Analysis** | FE-SEM image analysis | NMAM 0600  NMAM 7500 | Laser diffraction | Extra backup | Laser diffraction with optical analysis |
| **Purpose** | Particle shape | Concentration measurements  Mineralogy | Size distribution | Extra backup | Size distribution  Particle shape |

**Figure 5.** Representation of the instruments used for sampling, analyses performed, and characterization output obtained for the various particles collected.

The NIOSH Manual of Analytic Methods (NMAM) analyses are performed on the dust collected by a professional third party laboratory. The NMAM 0600 values provided are the dust concentrations normalized to the duration of collection time. For example, one test might have run for 11 min and another test run for 12 min with the pumps running at 1.7 L per minute, therefore, the concentration values are normalized and calculated by dividing the weight of dust collected by the volume of air that passed through the respective pump knowing the run time and pump flow rate. Additionally, the NMAM 7500 standard provides the micrograms of cristobalite, quartz, and tridymite detected in the respective collected samples. It is critical to perform these tests on the samples because

these are the standardized tests performed in the United States by industry and are used to regulate and mitigate dust exposures.

The third party laboratory followed the prescribed procedure for the NMAM 0600 and 7500 standards. For the gravimetric analysis in the NMAM 0600, the lab used a Mettler model XP6 balance with 0.001 mg accuracy. The media blanks used for the level of detection and limit of quantitation were provided from the laboratory as pre-weighed PVC filters from the same filter series. Additionally, some samples underwent a replicated analysis, where all the replicate results were within the 20% relative percentage difference limit.

To determine the particle shapes, a field emission scanning electron microscope (FE-SEM) and image analysis program were used to capture and process images from the PC filter surfaces. A Tescan FE-SEM at a voltage of 15 kV with back scattering electron (BSE) detection is used to obtain images. Then, the Clemex Vision PE® software processed the images, detected particles, and performed calculations of roundness, aspect ratio, and roughness of particles. The Clemex Vision PE® software was programmed to detect the particles with binarization by grey thresholding [42–44], which distinguish the difference between particles and the background. Other commands were also added to the program, such as bridging and object transfer, which separates particles that are clumped together and removes particles intersecting the edge of the image during detection, respectively.

## 3. Results from Analyzing the Dust Samples

Characterization of the respirable suspended limestone dust provided data for concentration, mineral composition, particle shape, and particle size distribution for the collected samples. Additionally, characterization of the limestone deposited material resulted in analyzing particle shape and particle size distribution.

### 3.1. NIOSH Manual of Analytical Methods

The NMAM 0600 results obtained from the multiple experiments are presented in Figure 6 as concentration. Concentration values reflect the unit of measurement used in industry for worker exposure values, such as time-weighted averages, ceiling values, or threshold limits. The data reveal that the concentration of the suspended respirable dust from the new to worn pick in mg/m$^3$ increases with increase in pick wear for all three test sets. For example, the respirable dust particle concentrations in test set two increased from 25 mg/m$^3$ to 68 mg/m$^3$ to 149 mg/m$^3$ as the wear of pick tip progressed from new to worn. Noting that the uncertainty, or precision, of the NMAM 0600 varies less than 10 μm for each sample with a 0.001 mg sensitivity balance, the trend between pick wears remains.

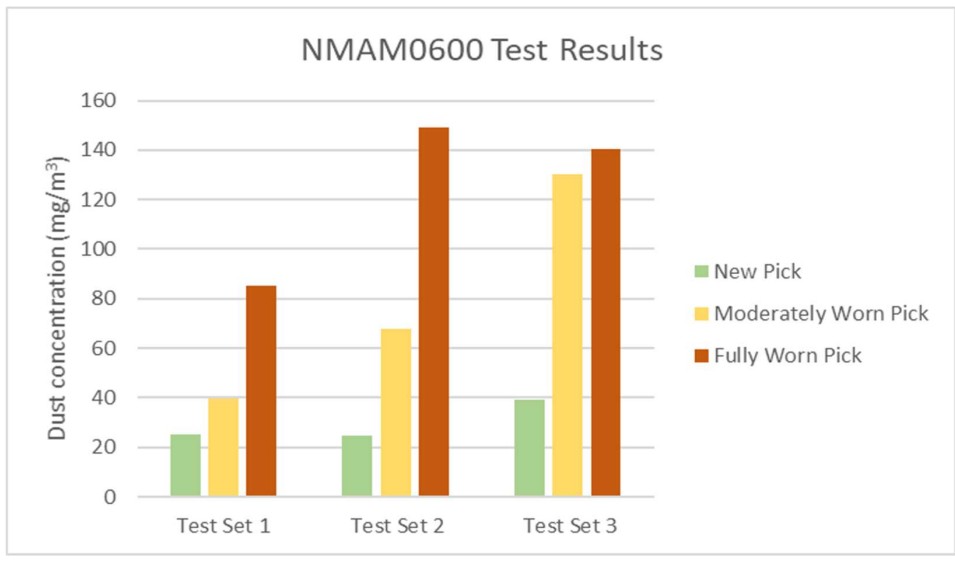

**Figure 6.** Concentration levels of suspended respirable limestone dust generated for each pick wear.

The generation of more dust particles from a worn pick compared to a new pick confirms findings in other experiments where a duller pick tip generated more dust [17]. This re-occurring trend is possibly due to the worn pick having more surface area contact with the rock compared to the new pick. In the end, it is recommended that more caution should be taken when cutting with worn picks compared to new picks because it is expected that more dust will be generated with worn picks. This could mean increasing the ventilation flows or incorporating water suppression systems.

The results from the NMAM 7500 standard reveal that all picks generated dust with silica containing minerals. Mainly quartz with a fraction of cristobalite were detected in all the samples, which are both considered hazardous minerals to human respiratory health. Although silica is present is all the samples, there is no apparent correlation between pick wear and silica concentration as seen in Figure 7. The quantitative differences of percent silica could result from the amount of dust collected per sample or nature of the grains in the limestone block. Therefore, it could be the limestone block itself which caused the difference in amount of quartz and cristobalite instead of the wear of the pick. It is inconclusive whether the change in amount of quartz between picks is due to the rock material, due to the changing pick tips, or due to the fact that the concentration of dust obtained per sample increased with pick wear which provided more or less material to detect.

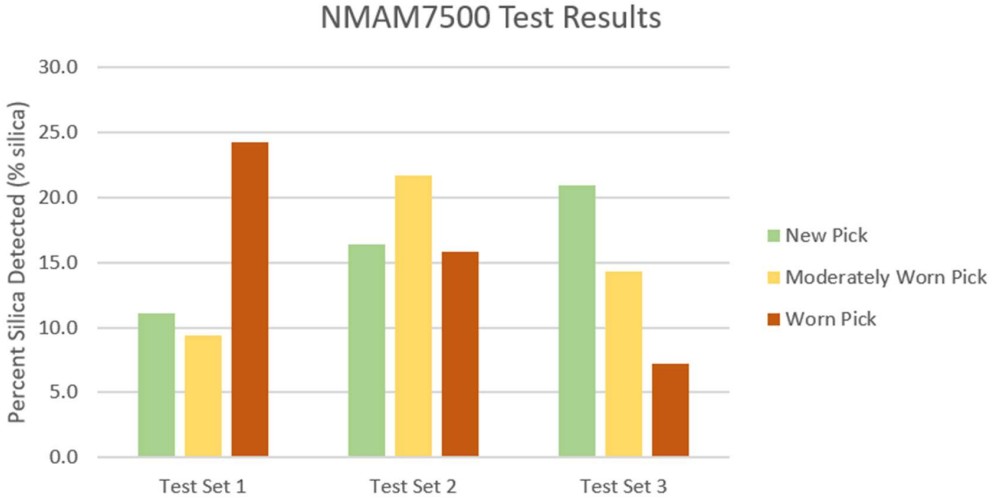

**Figure 7.** Percent silica detected per total sample of the suspended respirable limestone dust generated from each pick wear for the three test sets.

### 3.2. Particle Shapes

The roundness, aspect ratios, and roughness parameters were determined to obtain the general characteristics of the particle's shapes. With 20 images obtained per pick wear sample in the FE-SEM, at least 1040 individual particles were analyzed per sample. Therefore, the particle shape values in each row of Figure 8 belong to the roughly one-thousand particles analyzed from one sample. Additionally, over 103,000 individual particles were detected for each sample and analyzed optically from the laser diffraction analysis. Therefore, the values in each row in Figure 9 belong to the same individual sample.

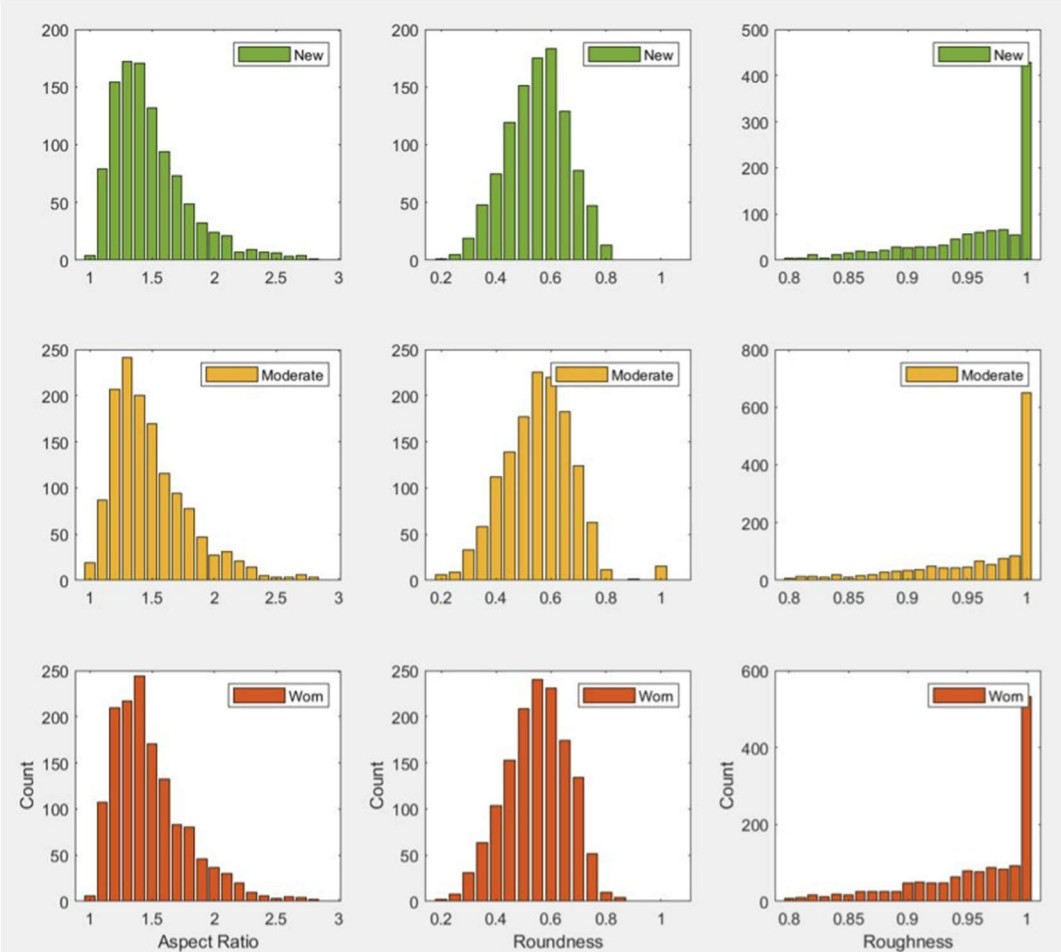

**Figure 8.** The aspect ratio, roundness, and roughness values calculated and represented in histograms for the suspended respirable particles generated from the three different pick wears.

For both the FE-SEM and laser diffraction image analyses, silica and calcium carbonate particles were mixed. Therefore, using Figure 7, it can be presumed that about 15% of the particles analyzed are silica and 85% of the particles analyzed are calcium carbonate. However, lacking a formal energy dispersive X-ray spectroscopy analysis, it is indeterminate if there is a difference between particle shapes of the various elemental compositions.

Particle roundness is calculated to determine how close to a perfect circle the particle of interest is via Equation (1):

$$\text{Roundness} = \frac{4\,(\text{area})}{\pi\,(\text{length})^2} \tag{1}$$

Particle aspect ratio is calculated to determine the elongation of particles via Equation (2):

$$\text{Aspect Ratio} = \frac{\text{length}}{\text{width}} \tag{2}$$

Particle roughness is calculated to determine the smoothness of the particle perimeter via Equation (3):

$$\text{Roughness} = \frac{\text{ConvexPerimeter}}{\text{Perimeter}} \tag{3}$$

Convex perimeter is used to determine the particle roughness, where the convex perimeter is the perimeter of the object if a rubber band were placed around the particle as shown in Equation (4):

$$\text{ConvexPerimeter} = \sum \text{ferets} \left[ 2 \tan \left( \frac{\pi}{2(\text{number of ferets})} \right) \right] \tag{4}$$

The length is the longest measurement across an object and the width is the shortest distance measured across an object. A feret is the distance between two parallel tangents on each side of an object in the image, which is analogous to using a caliper.

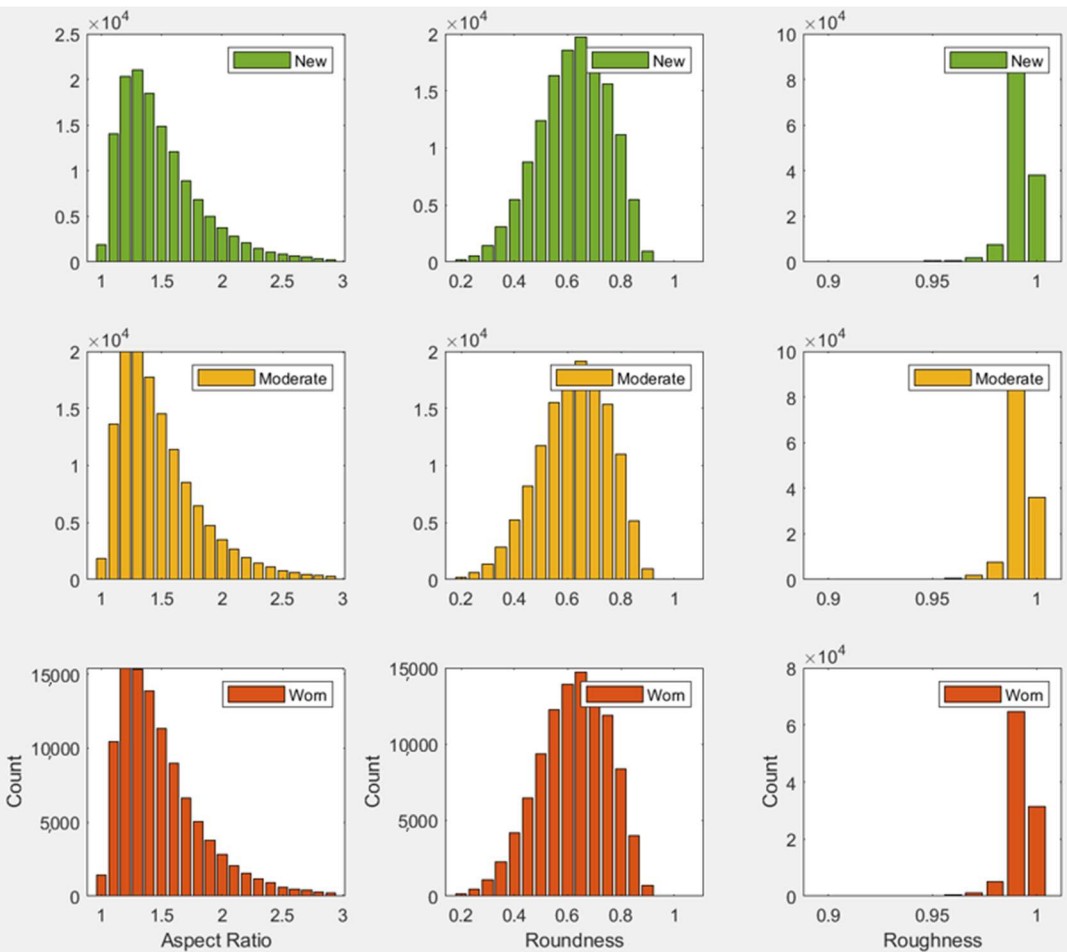

**Figure 9.** The aspect ratio, roundness, and roughness values calculated and represented in histograms for the deposited particles generated from the three different pick wears.

Figure 8 shows histograms that visualize the number of particles with specific values for the roundness, aspect ratios, and roughness in relation to their particle counts.

The results obtained on the roundness, aspect ratio, and roughness measures of respirable particles show that there are few statistically significant differences between the shapes generated by various picks. The Kolmogorov-Smirnov statistical test (KS test) was performed between each of the datasets to test the null hypothesis that the distributions of particle shapes generated from the picks are the same. A significance level of 0.05 is used in the statistical analysis meaning that *p*-values below 0.05 provide strong statistical evidence to reject the null hypothesis.

Visually observing Figure 8 reveals that the particle shapes are similar between picks at different levels of wears. The KS test fails to provide strong evidence that respirable dust generated by the new, moderately worn, and fully worn picks have significantly different

roundness and aspect ratio measures, as all the *p*-values are 0.34 or larger. However, the statistical test suggests that the moderately worn pick generates dust with different roughness measures compared to the new and fully worn picks. The p-value obtained for the KS test was 0.0009 between the new and moderately worn pick, and $4.6 \times 10^{-7}$ between the moderately worn and worn picks for roughness measures. The observed differences in roughness between the picks may be due to random variation in the limestone sample rather than the differences in wear between picks. Aside from these limited values, there is weak statistical evidence that all the picks generate respirable dust particles of different shapes.

Concluding that the respirable particle shapes stay the same no matter which pick wear generates dust, the respirable limestone particles obtained the following particle shape characteristics. The average aspect ratio was 1.5, the average roundness value was 0.58, and the average roughness value was 0.96.

The particle shapes were also analyzed for the deposited particles. Although a few of the nine total *p*-values calculated from KS tests for the deposited particle datasets were also below the 0.05 threshold for shape measurements, the histograms in Figure 9 provide strong evidence that the deposited particle shapes generated are all also similar between the three pick wear conditions. Overall, there is weak statistical and graphical evidence that the deposited dust particle roundness, aspect ratio, and roughness shapes generated from all the picks are significantly different.

Concluding that the particle shapes stay the same irrespective of pick wear, the deposited particles obtained the following particle shape characteristics. The average aspect ratio was 1.6, the average roundness value was 0.65, and the average roughness value was 1.00.

### 3.3. Particle Size Distributions

Laser diffraction was used to determine the size distribution of the respirable suspended and deposited particles. The size distributions for suspended respirable particles are shown in Figure 10. The suspended respirable particle size distributions were also calculated from the FE-SEM image analysis, which confirms the curves in Figure 10 with the smallest particles generated by the worn pick, the largest particles generated by the new pick, and the moderately worn pick generating particle sizes between the two. Additionally, the size distributions for the collected deposited particles from the cuts are presented in Figure 11.

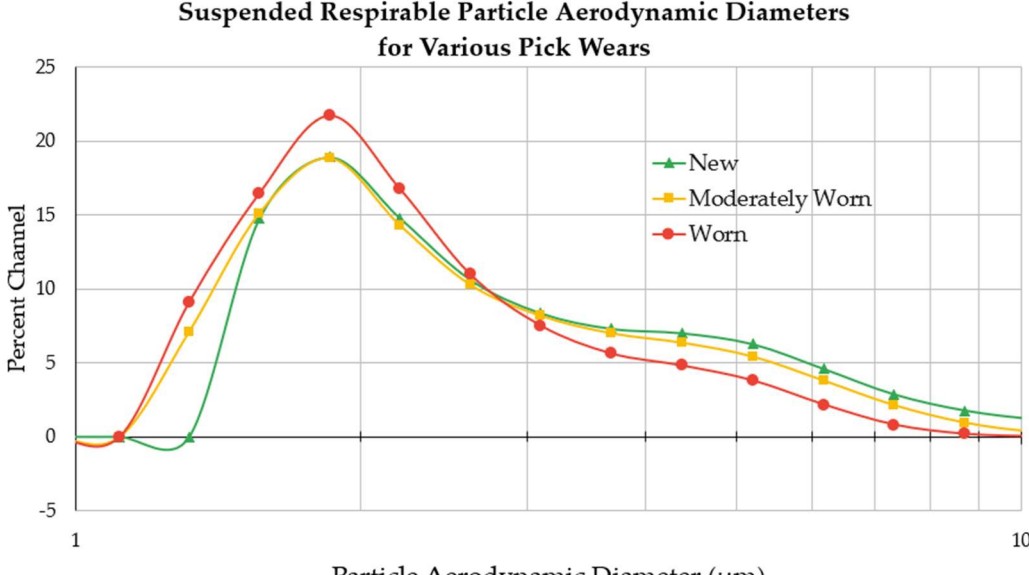

**Figure 10.** The particle size distribution of the suspended respirable particles generated from the new, moderately worn, and fully worn pick determined from laser diffraction of particles collected onto PC filters with cyclones.

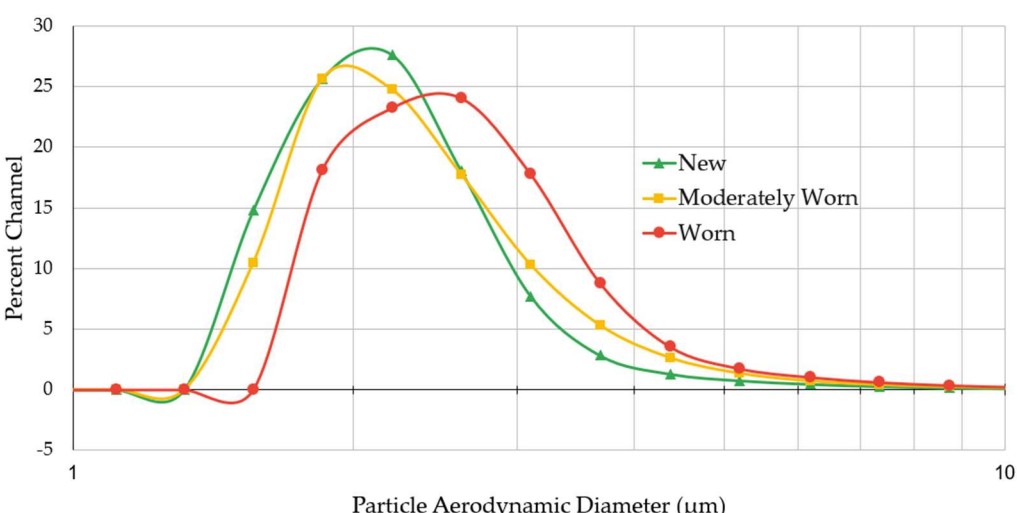

**Figure 11.** The particle size distribution of the deposited particles generated from the new, moderately worn, and fully worn pick determined from laser diffraction of particles collected from the surface of the rock block.

Particle count is used in the particle size analysis where the "percent channel" in the laser diffraction analysis counts each particle and places them in bins depending on the size. The aerodynamic diameters of the particles were calculated from the physical diameters using the Cunningham correction factor and slip correction factor [45]. Additionally, each colored curved displayed is the average of three duplicate sample runs in both Figures 10 and 11.

The results obtained from the suspended respirable particles and the deposited particles show that the picks generated slightly different size distributions. KS statistical tests were performed between each of the datasets to test the null hypothesis that the particle size distributions generated from the picks are the same. A significance level of 0.05 was used, meaning that *p*-values below 0.05 provide strong statistical evidence to reject the null hypothesis. All *p*-values between datasets for both respirable and deposited particles were beneath the significance level. Therefore, there is weak statistical evidence that the picks generated suspended respirable and deposited dust particles of similar size distributions.

As seen in Figures 10 and 11, each pick generated particle size distributions with a similar single modal crest curve. However, in the suspended respirable analysis there is a shift, or trend, in the curves where the new pick generated larger particles, the worn pick generated smaller particles, and the moderately worn pick generated particle sizes between the two. Additionally in the deposited particle analysis, there is a shift, or trend, in the curves where the new pick generated smaller particles, the worn pick generated the larger particles, and the moderately worn pick generated particle sizes between the two.

It is uncertain why the new pick generated larger suspended respirable particles, the worn pick generated smaller particles, and the worn pick generated particle sizes between the two. However, the slightly smaller deposited particle sizes generated from the new pick could possibly be attributed to the smaller surface contact area and point force that occurs during rock cutting with newer picks [46–49]. The single, smaller point of contact could lead to further crushing of the rock in this zone due to higher force concentration or higher stresses in the pressure bubble under the pick tip. In the end, there is a shift between the particle sizes on the suspended respirable and deposited particle graphs, but the shifts are not significant amounts, as most particle sizes generated from all the picks are still in the respirable region between 1.2μm and 4μm in aerodynamic diameter.

## 4. Discussion

All picks generated dust containing silica-bearing minerals such as quartz, with traces of cristobalite. However, the amount of quartz and cristobalite generated is believed to derive from the mineralogy of the rock and not necessarily the pick configuration. Additionally, the concentration of dust increases with the increase in pick wear, as represented in this study by the radius of the pick tip nose. These findings support previous studies where coal samples were cut with various pick types and wear conditions and an increase in dust concentration was noted with the increase in pick wear [17,19,22]. Therefore, when drilling limestone with conical picks, appropriate engineering dust mitigation measures should be considered to protect workers at all levels of pick wear. Additionally, dust suppression measures will possibly need to be increased as mining continues with the same cutting tools on the excavation unit because as picks wear progresses, there is evidence that dust concentration increases.

All picks generated similar dust particle shapes and similar particle size distributions as seen in Figures 8, 9 and 12. Therefore, when drilling limestone with conical picks, appropriate engineering dust mitigation measures should be considered to protect workers at all levels of pick wear.

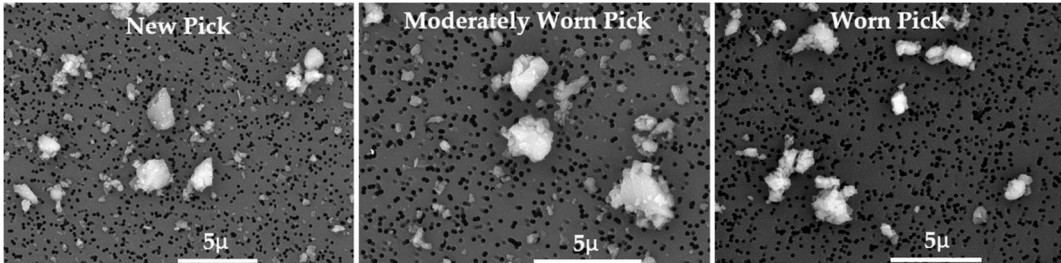

**Figure 12.** Raw FE-SEM example images of the particles generated from the new, moderately worn, and fully worn pick that shows the particles being similar shapes and sizes.

The picks generated suspended respirable particle size distributions in which smaller particles were generated as the pick wear increased. The picks generated deposited particle size distributions in which larger particles were generated as the pick wear increased. Although there is a shift and trend present in the particle size distributions for suspended and deposited particles, all the particle size distributions are within the respirable size range. Therefore, it is necessary to provide dust mitigation methods while mining limestone with all pick wear levels to mitigate workers inhaling the particles that can penetrate deep into their lungs.

The findings support previous preliminary studies regarding concentration and respirable particle size distributions [50,51]. Although previous studies used only two pick wears to cut an igneous rock, by using three pick wears in the limestone cutting tests, the trend of increasing concentration as the pick wears is consistent. Additionally, the size distributions of the limestone suspended respirable particles agree with previous studies as there are smaller particle sizes with the increase in pick wear. Substantial improved elements in the cutting tests on limestone include utilizing FE-SEM to observe the particle shapes and collect a larger dataset of dust particles. With the increase in resolution and ability to look at smaller particles, the limestone cutting tests contain more accurate results of the particle shape analysis compared to the optical analysis of particle shapes in previous preliminary studies.

In the end, the characterization results obtained from cutting limestone with various pick wear conditions can be used in the selection of proper dust suppression systems for the underground environment where mechanical excavation is used. Further research needs to be completed on other coal, metal and non-metal rock types to confirm the trends and comparisons drawn from the results. Planned experiments using similar methods and analysis are underway with a sandstone sample. Additionally, further research on the

effects of particle shapes and sizes on the human respiratory system is necessary. With this information, more appropriate dust mitigation measures could be instated relating to the specific aspect ratios, roundness, and roughness measures in relation to the sizes of the limestone particles.

## 5. Conclusions

Full-scale cutting tests of a limestone sample with a new, moderately worn, and worn conical pick were used to generate dust for analysis presented in this study. Characterization of the suspended respirable dust from all three wear conditions resulted in obtaining and comparing data for concentration, mineral presence, particle shape, and particle size distribution. Additionally, characterization of the deposited particles generated from all three wear conditions resulted in analyzing particle shape and particle size distribution. The worn pick generated the most overall suspended respirable dust and all three picks generated dust containing quartz with traces of cristobalite. In general, all the picks generated suspended respirable and deposited particles with similar particle shapes in terms of roundness, aspect ratio, and roughness measures. The suspended respirable particle size distributions reveal that as the pick wear increases, the particle sizes slightly decrease. The deposited respirable particle size distributions reveal that as the pick wear increases, the particle size slightly increases. Although there are trends in the shifts of particle size distributions, all the particle sizes are within the respirable range and should be controlled because they pose a threat to human health.

**Author Contributions:** Conceptualization, S.S. and J.R.; methodology, S.S. and J.R.; software, S.S.; validation, J.R., J.B. and C.T.; formal analysis, E.S.; investigation, S.S.; resources, S.S., J.R., J.B. and C.T.; data curation, S.S. and E.S.; writing—original draft preparation, S.S.; writing—review and editing, J.R., J.B., C.T., E.S.; visualization, S.S. and E.S.; supervision, J.B.; project administration, J.R.; funding acquisition, J.R. All authors have read and agreed to the published version of the manuscript.

**Funding:** This research was funded by the National Institute of Occupational Safety and Health and the Center for Disease Control and Prevention (NIOSH/CDC), grant number 75D30119C05413 (Improving Health and Safety of Mining Operations Through Development of the Smart Bit Concept for Automation of Mechanical Rock Excavation Units and Dust Mitigation).

**Institutional Review Board Statement:** Not applicable.

**Informed Consent Statement:** Not applicable.

**Data Availability Statement:** Data supporting reported results can be provided upon request. Please contact the corresponding author for data.

**Acknowledgments:** Authors would also like to acknowledge various individuals for their guidance and contributions, which includes Muthu Vinayak Thyagarajan, Brent Duncan, Michelle Reiher, Hannah Rognerud, Jeremy Pacheco, and Tom Lillis.

**Conflicts of Interest:** The authors declare no conflict of interest. The funders had no role in the design of the study; in the collection, analyses, or interpretation of data; in the writing of the manuscript, or in the decision to publish the results. The funders only provided the following equipment: the Dorr-Oliver cyclones, air pumps, and filter media for collecting the dust.

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
