# Peer review of "Characterization of Respirable Dust Generated from Full Scale Cutting Tests in Limestone with Conical Picks at Three Stages of Wear"

_minerals, doi:10.3390/min12080930_

Round 1

Reviewer 1 Report

An interesting research is provided by authors in order to occupational exposure in workplace.

Nevertheless I believe that the same paper has many weaknesses that has to be overcome. Authors did not consider the background due to emissions of environmental sources compare with the emission of limestone dust. Moreover in the paper has not been calculated the measurement uncertainty in the gravimetric data sets. Finally, the work doesn’t show new substantial elements comparing to the works previously published by the same authors[1-2]. With regard to the above I cannot recommend this paper for publication in Minerals journal

[1] Slouka S; Rostami J; Brune J. Characterization of respirable dust samples generated from picks at differing stages of wear. Mine Ventilation: Proceedings of the 18th North American Mine Ventilation Symposium (NAMVS 2021), June 12-17, 2021, Rapid City, South Dakota. Tukkaraja P ed. London: CRC Press, 2021 Jun; :198-207. https://doi.org/10.1201/9781003188476-20

[2] Slouka, S., Brune, J. & Rostami. Characterization of Respirable Dust Generated from Full-Scale Laboratory Igneous Rock Cutting Tests with Conical Picks at Two Stages of Wear. Mining, Metallurgy & Exploration, 2022. https://doi.org/10.1007/s42461-022-00625-w

Reviewer 2 Report

The paper deals with a problem that is undoubtedly of relevance to the health aspects of mining and engineering activities. The results are well presented and the statistical data treatment is sound. The paper can be accepted for publication, but it would benefit greatly from a  substantial revision. The authors should pay more attention to the language, because the manuscript contains more than a few crooked sentences. 

In general the methodology and the measurement equipment is well described. However, something very important is missing : a description of the nature of the limestone test material. A rock is characterized by more than just four petrophysical parameters. Now, the discussion of the mineralogy and petrography pops up late in the paper, because of the presence of silica (quartz, cristobalite) particles. However, the presence of these carcinogenic particles is of little relevance to the main topic of the paper : the effect of increasing pick wear. It is just a result of the (accessory) mineralogy of the test rock. Figure 7 should be better placed in the  ‘materials and methods’ section of the paper. Fig.7 shows that the test sample was a clastic , micritic limestone. One cannot preclude that cutting of coarse-grained sparitic limestone  (or silicate rocks for that matter) would produce significantly different populations of respirable dust particles. 

A highly questionable aspect of the paper is the distinction between ‘airborne particles’ and ‘fines material’.  It is unlikely that all the airborne particles have been captured by the various suction/filter devices. The collection effciency must have depended on the dimensions of the hood around the pick and on the air flow characteristics inside it.  The largest fraction of the ‘fines material’ must have originally been airborne as well. Actually, this is what the results clearly show.

One can argue that it is more appropriate to speak of ‘suspended particles’ and ‘deposited particles’. This is just an operational distinction for the given experimental setup. One could expect that the deposited material contained a higher proportion of somewhat larger particles that have a shorter residence time in the air around the pick. This could explain the shift in peak values in Figures 10 and 11. Admittedly, it offers no explanation for the hump between 4 and 7 µm in the distribution of ‘respirable’ particles (Figure 10).

Some information that needs to be clarified : 

- lines 91-94 : This sentence does not state unambiguously what a ‘pass’ actually is.- also : how is ‘test set’ defined ? 

- lines 218-219 : ‘25 mg/m3 , etc.’     What does cubic meter mean in this context ? 

-lines 228-237 : This section is very confusing . Are the amounts of silica particles normalized to the amounts of collected dust ?  If not, why not ?

Judging from Figure 7, the silica minerals are only 20 µm in diameter.  It is unlikely that an completely new pick with a tip diameter of ca. 3500 µm (see line 169) would produce less silica particles smaller than 10 µm than a worn pick. 

- line 201 : Did one really use Back Scattered Electron images for particle analysis ? 

Some editorial remarks : 

- line 16 : The work is relevant for other activities than mining . Better phrasing : “ …. the lifetime of a pick in underground mining and engineering activities.”  

- lines 22-23 : ‘various particle size distributions with trends’ : this sentence is completely incomprehensible outside its context. By the way, what is ‘trend’ meaning here ? (see also later in de paper). 

-line 125 : It is standard practice to write  ‘US 200 Mesh (<74µm)’ . 

-lines 121 and 174 : ‘shop vac’ and ‘Dremel’ are brand names, and should be marked with the TM sign (as done for Tygon in line 137). 

- lines 150 and 159 : One can only do one thing ‘Lastly’. 

- line 266 : What is ‘.. relative to the processing frame..’ ?? 

- lines 46, 173,228, 265… : Sentences starting with “With this,…”. This is informal coffee break English with little legitimacy in scientific papers. 

- caption Figure 2 : use capitals (A),(B), etc. for consistency with the figure. 

- caption Figure 7 : What is implied by ‘compacted grains’ ?. In standard petrographic terminology ‘compacted’ means ‘flattened under directional pressure’. This is not the case in this picture. 

- Figures 10 and 11 : typo in x-axis legend : ‘Paricle’ instead of ‘Particle’ 

- line 343 : ‘with the moderately worn pick generating….‘

Round 2

Reviewer 1 Report

The authors have resolved most of the issues highlighted in the first version. I think that the manuscript still lacks originality, it would be interesting if the authors could indicate a pick wear time for different rock types in relation to particle shape and size to improve dust mitigation measures, but for this I think that additional experiments needed

There are some errors as noted below:

1)    In line 67 is indicated non-mental I believe it should be non-metal

2)    In line 88 is indicated 4.4 sec, from my calculations results 4.2 sec

In figure 8 it would be appropriate to have the same y-axis values for the roughness graph
